# ZERO-SHOT FAIRNESS
# WITH INVISIBLE DEMOGRAPHICS

## ABSTRACT

In a statistical notion of algorithmic fairness, we partition individuals into groups based on some key demographic factors such as race and gender, and require that some statistics of a classifier be approximately equalized across those groups. Current approaches require complete annotations for demographic factors, or focus on an abstract worst-off group rather than demographic groups. In this paper, we consider the setting where the demographic factors are only partially available. For example, we have training examples for white-skinned and dark-skinned males, and white-skinned females, but we have *zero examples* for dark-skinned females. We could also have zero examples for females regardless of their skin colors. Without additional knowledge, it is impossible to directly control the discrepancy of the classifier's statistics for those invisible groups. We develop a disentanglement algorithm that splits a representation of data into a component that captures the demographic factors and another component that is invariant to them based on a context dataset. The context dataset is much like the deployment dataset, it is unlabeled but it contains individuals from all demographics including the invisible. We cluster the context set, equalize the cluster size to form a "perfect batch", and use it as a supervision signal for the disentanglement. We propose a new discriminator loss based on a learnable attention mechanism to distinguish a perfect batch from a non-perfect one. We evaluate our approach on standard classification benchmarks and show that it is indeed possible to protect invisible demographics.

## 1 INTRODUCTION

Machine learning is already involved in decision-making processes that affect peoples' lives such as in screening job candidates (Raghavan et al., 2020) and in pricing credit (Hurley & Adebayo, 2017). Efficiency can be improved, costs can be reduced, and personalization of services and products can be greatly enhanced – these are some of the drivers for the widespread development and deployment of machine learning algorithms. Algorithms such as classifiers, however, are trained from large amount of labeled data, and can therefore encode and even reinforce past discriminatory practices that are present in the data. The classifier might treat some groups of individuals unfavorably, for example, denying credit on the grounds of language, gender, age and their combined effect. Algorithmic fairness aims at building machine learning algorithms that can take biased datasets and outputs fair/unbiased decisions for people with differing protected attributes, such as race, gender, and age.

A typical setting of algorithmic fairness is as follows. We are given a training set of observations $x \in \mathcal{X}$, their corresponding protected attributes $s \in \mathcal{S}$, and the target label $y \in \mathcal{Y}$ for learning a classifier. In a statistical notion of algorithmic fairness e.g. (Kamiran & Calders, 2012a; Hardt et al., 2016; Zafar et al., 2017), we control the discrepancy of a classifier's loss for a small number of demographic groups defined on protected attributes. Recently, several works have considered the setting where protected attributes are unknown (Kearns et al., 2018; Hashimoto et al., 2018; Khani et al., 2019). They aim to control the losses of groups whose size is greater than some predefined value. These works focus on an *abstract* worst-off group rather than demographic groups. It has been noted that the implied worst-off groups may differ from well-specified demographic groups who are known to suffer from past discriminatory practices (Hashimoto et al., 2018).

We are interested in the setting that is in between having complete annotations for demographic groups and having none. In this paper, we introduce algorithmic fairness with *invisible demographics*.

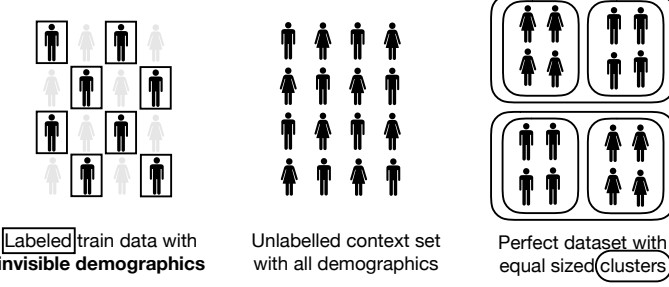

Figure 1: Overview of learning with invisible demographics: the train dataset, the context dataset and the proposed approach using a perfect dataset.

Who are the invisible demographics? In the context of machine learning systems, those are individuals with thin or non-existent labeled training data. The invisible population is primarily composed of individuals with certain protected attributes (Hendricks, 2005; Abualghaib et al., 2019; Perez, 2019). We now elaborate on several algorithmic decision scenarios involving invisible demographics. One scenario is when we observe partial outcomes for some of the demographic groups, e.g. we have labeled training data for males (with positive and negative outcomes), but for the group of females, we only observe the one-sided labels (negative outcome). Another scenario is when we do not observe any outcome for some of the demographic (sub)groups, e.g. we have training samples for white-skinned and dark-skinned males, and white-skinned females, but we have zero labeled data for dark-skinned females. An extreme version of the last scenario is when we do not observe any outcome for females regardless of their skin colors, e.g. we only have training samples for males and no training examples for females. To summarize, in the invisible demographics problem, we *define* the demographics groups that are expected to be seen, so they are not abstract. However, not all of the demographics are observed (labeled) during training, forming missing or invisible demographics.

This paper presents learning disentangled representations in the presence of invisible demographics. Our source of supervision is motivated by the observation that we want to deploy our classifier to the eventual real-world population. This deployment dataset will contain individuals from all demographics. We thus consider the setting where unlabeled data is available for learning disentangled representation. We call this data a *context set* and this context set is much like the deployment dataset, it is unlabeled but it contains all demographics including the invisible ones.

We aim to convert our unlabeled context set into a *perfect dataset* (Kleinberg et al., 2016; Chouldechova, 2017), a dataset in which the target label and protected attribute are independent (i.e. $y \perp s$). We will then use this perfect dataset as the inductive bias for learning disentangled representations. How do we construct this perfect dataset without labels? We assume that the number of demographic groups (hence clusters) is known a priori corresponding to the diverse demographic groups in the real-world population in which our machine learning system will be deployed. We use unsupervised kmeans clustering, or a supervised clustering based on rank statistics; the latter one allows to form the clusters that also support annotations in the train data. Once the clusters have been found, we can equalize the cluster size to form a perfect dataset and use it as an input for learning a disentangled fair representation. See fig. 1 for an overview of our learning with invisible demographic framework.

Specifically, our paper provides the following main contributions:

1. A problem of algorithmic fairness with invisible demographics where we have *zero data* for some of demographics and we still have to make predictions for those groups.
2. Applying clustering methods to the task of transforming *unlabeled context dataset* into a perfect dataset.
3. Theoretical and experimental justification that the disentangled model with *the perfect dataset as an inductive bias* provides a well-disentangled fair representation, one component captures the demographic factors and another component is invariant to them.

**Related work** We describe related work in three areas: zero-shot learning, semi-supervised learning, and disentangled representation learning. **On zero-shot learning.** The setting with incomplete training data, where we aim to account for seen and unseen outcomes is also known as *generalized zero-shot learning*. Traditionally, zero-shot learning transfers knowledge from classes for which we

have training data to classes for which we do not have via auxiliary knowledge, e.g. via prototype examples (Larochelle et al., 2008), intermediate class description such as semantic attributes (Lampert et al., 2009; Xian et al., 2018), word2vec embeddings (Bucher et al., 2019). Our method similarly uses a context set as a source of auxiliary knowledge but in in contrast to generalized zero-shot learning, our context set is an unlabeled pool of data, where class descriptions are unknown. **On semi-supervised learning.** Wick et al. (2019) proposed a semi-supervised method that can successfully harness unlabeled data to correct for the selection bias and label bias in the training data. The unlabeled data, despite not containing the target label $y$, *is* labeled in terms of the protected variable $s$. Our setting is significantly harder because there is no label information about $y$ and $s$ in the context set. **On disentangled representations learning.** Locatello et al. (2019a) suggested that disentanglement in representation learning may be a useful property to encourage fairness when protected variables are not observed. In order for disentangled representations to improve fairness measure without the knowledge of protected attribute $s$, they have to assume that the target label $y$ and the protected attribute $s$ are independent, i.e. $y \perp s$. Though, in fairness settings, the variable $s$ is correlated with the variable $y$, and therefore unsupervised methods are not suitable for fairness (Jaiswal et al., 2018b; 2019). Indeed, experiments in (Locatello et al., 2019a) were wholly done with procedurally generated synthetic datasets involving 2D and 3D shapes. Without some supervision or inductive bias, disentangled representation methods would not solve the issue of algorithmic fairness with invisible demographics (Locatello et al., 2019b).

## 2 METHODOLOGY

### 2.1 THEORETICAL BACKGROUND

In this section, we first formulate mathematically the problem of invisible demographics and its associated issue of algorithmic fairness. We then motivate theoretically the idea of perfect dataset for achieving fairness, and its use for an inductive bias in learning disentangled representations.

**Invisible demographics and algorithmic fairness.** Let $S$ denote the set of discrete-valued protected attributes with an associated domains $\mathcal{S}$. $S$ can take the values taken by a single protected attribute, or, $S = S_1 \times S_2 \times \ldots \times S_p$ with $S_1, \ldots, S_p$ be discrete-valued protected attributes more generally. $X$, with the associated domain $\mathcal{X}$, represents other attributes of the data. Let $\mathcal{Y}$ denote the space of class labels for a classification task ($\mathcal{Y} = \{0, 1\}$ for binary classification or $\mathcal{Y} = \{1, 2, \ldots, C_{\text{cls}}\}$ for multi-class classification). For ease of exposition, we assume that we have multiple sources $\mathcal{M}$ of samples, one for each combination of class label $y$ and protected attribute $s$. That is, we have:

$$\mathcal{M}_{ys}, \qquad \forall y \in \mathcal{Y}, \forall s \in \mathcal{S}, \tag{1}$$

where, for example, the source $\mathcal{M}_{y=0,s=0}$ contains all data points with class label $y = 0$ and protected attribute $s = 0$. As in a standard supervised learning task, we have access to a training set $D^{\text{tr}} = \{(x_i, s_i, y_i)\}$, that is used to learn a model $M : \mathcal{X} \to \mathcal{Y}$. $D^{\text{tr}}$ is composed of several sources. This labeled training dataset, however, lacks samples from some of the sources:

$$\exists y \in \mathcal{Y}, \exists s \in \mathcal{S} : D^{tr} \cap \mathcal{M}_{ys} = \varnothing. \tag{2}$$

For example, we might not have samples from two sources: $\mathcal{M}_{y=0,s=0}$ and $\mathcal{M}_{y=1,s=0}$. In binary classification, this corresponds to zero-labeled data for the *invisible demographic* group $s = 0$. Or we only observe a negative outcome for the invisible demographic $s = 0$, i.e. we have $\mathcal{M}_{y=1,s=0} = \varnothing$.

Once the model $M$ is trained, we deploy it to the real-world population with diverse demographic groups. That is, we have a deployment set, $D^t = \{(x_i)\}$ which has overlap with all sources:

$$D^t \cap \mathcal{M}_{ys} \neq \varnothing \qquad \forall y \in \mathcal{Y}, \forall s \in \mathcal{S}. \tag{3}$$

If the model relies only on the incomplete training set, it is not unreasonable to expect that the model to easily misunderstand the *invisible*s. We can all agree that this sounds unfair, and we would like to rectify this. We will be precise shortly about the adopted mathematical definitions of fairness.

We propose to alleviate the issue of unfairness to the *invisibles* by mixing labeled with unlabeled data, which is usually much cheaper to obtain. In this paper, we call this unlabeled data a context set $\mathcal{D}^{ctx} = \{(x_i)\}$. This context set has overlap with all sources:

$$\mathcal{D}^{ctx} \cap \mathcal{M}_{ys} \neq \varnothing \qquad \forall y \in \mathcal{Y}, \forall s \in \mathcal{S} \tag{4}$$

The context set is much like the deployment set: it has no information about class labels $y$ or the protected attributes $s$.

We adopt a statistical notion of algorithmic fairness in which it balances a certain condition between groups of individuals with different protected attributes. The term $\bar{y}$ below is the prediction of a machine learning model $M$. Several statistical fairness criteria have been proposed (Kamiran & Calders, 2012a; Hardt et al., 2016; Zafar et al., 2017; Chouldechova, 2017; Raghavan et al., 2020) (shown below for the case where $s$ and $y$ are binary):

$$\Pr(\bar{y} = 1|s = 0) = \Pr(\bar{y} = 1|s = 1) \qquad \text{(equality of acceptance rate)} \quad (5)$$
$$\Pr(\bar{y} = 1|s = 0, y) = \Pr(\bar{y} = 1|s = 1, y) \qquad \text{(equality of true positive/negative rate)} \quad (6)$$
$$\Pr(y = 1|s = 0, \bar{y}) = \Pr(y = 1|s = 1, \bar{y}) \qquad \text{(equality of positive/negative predicted value)} \quad (7)$$

Generally, those statistical notions can be expressed in terms of different (conditional) independence statements between the involved random variables (Barocas et al., 2019): $\bar{y} \perp s$ (equation 5), $\bar{y} \perp s \mid y$ (equation 6), and $y \perp s \mid \bar{y}$ (equation 7). If our training set has no positive outcome for the demographic $s = 0$, i.e. $\mathcal{M}_{y=1,s=0} = \varnothing$, the true positive rate for this group will suffer, and therefore we will likely not be able to satisfy, among others, equality of true positive rate.

**Perfect dataset.** We call a dataset for which $y \perp s$ holds, a perfect dataset (Chouldechova, 2017; Kleinberg et al., 2016). If we have access to a perfect dataset, we could equalize true positive/negative rates (eq. 6) and also equalize positive/negative predicted values (eq. 7) for all demographic groups. This can be shown by using the sum and product rule of conditional probabilities, e.g. (Kannan et al., 2019). Let's consider a binary-valued protected attribute, $s'$ versus $s''$. For $s'$, we can compute: $\Pr(y = 1|\bar{y} = 1, s') = \Pr(\bar{y}=1|y=1,s')\Pr(y=1|s')\big/\big(\Pr(\bar{y}=1|y=1,s')\Pr(y=1|s')+\Pr(\bar{y}=1|y=0,s')(1-\Pr(y=1|s'))\big)$, and accordingly for $s''$. The conditional probability on the left hand side is a positive predicted value, and this quantity can be expressed in terms of true positive/negative rates and the base (prior) rate, shown on the right hand side. If we have a perfect dataset ($y \perp s$ holds, which means equal base rates $\Pr(y = 1|s') = \Pr(y = 1|s'')$), an equality in the true positive/negative rates will give us an equality in the positive/negative predicted values. Similarly, with a perfect dataset, we can equalize true positive/negative rates (eq. 6) and also acceptance rates (eq. 5) for all demographic groups. From the sum probability rule, we have: $\Pr(\bar{y} = 1|s') = \Pr(\bar{y} = 1|y = 1, s')\Pr(y = 1|s') + \Pr(\bar{y} = 1|y = 0, s')(1 - \Pr(y = 1|s'))$ for $s'$ value, and accordingly for $s''$ value. Here, an acceptance rate on the left hand side is related to true positive/negative rates and the base (prior) rate as shown on the right hand side. In general, however, our given dataset is likely to be imperfect. In this paper, we pursue learning a fair classifier for all demographics as learning disentangled representations with an *approximately* perfect dataset.

**Disentangled representation.** Disentanglement learning aims to find a split representation of a data point $x$ and a mapping function $f$ such that $f(x) = (z_1, z_2, \ldots, z_p)$ where $z_1, z_2, \ldots, z_p$ are $p$ distinct (independent) factors of variations. We can mathematically formalize this intuitive definition using group and representation theories Higgins et al. (2018), or using structural causal models Suter et al. (2019). Specifically in this paper, we would like to split representation of data into two factors as $f(x) = (z_y, z_s)$ where $z_y$ contains factors that are relevant for $y$-prediction and $z_s$ contains factors related to demographic group $s$. As noted by Jaiswal et al. (2018a; 2019) (also vide sec. 1), since the protected variable $s$ is correlated with the class label $y$, we need annotations of undesired nuisance variable $s$ to be successful in using disentanglement learning methods for fairness. We only have annotations of variable $s$ in the training set $\mathcal{D}^{tr} = \{(x_i, s_i, y_i)\}$, however, crucially, this set contains missing demographic groups. We have all demographic groups in the context set $\mathcal{D}^{ctx} = \{(x_i)\}$ (also in the deployment set $\mathcal{D}^{tr} = \{(x_i)\}$), though, the challenge is we should not expect annotations of protected variable $s$ at the deployment time. The next section will show that we can still leverage the context set for learning the disentangled representations.

**Disentanglement with a perfect dataset.** Our framework for learning the disentangled representations comprises four core modules: 1) an *encoder* function $f$ that embeds $x$ into a bipartite space $f(x) \rightarrow (z_y, z_s)$; 2) a *decoder* function $g$ that learns the inverse of $f$, mapping back from the embedded space into the input domain $g(z_y, z_s) \rightarrow \tilde{x}$; 3) a *predictor* function $l$ that predicts $y$ from $z_y$, and 4) a *discriminator* function $h$ that classifies whether a given batch of samples embedded in $z_y$ derives from the either context set or the training set; this marks a significant departure from the typical GAN discriminator, which takes as input batches of data and yields a prediction for each sample independently of the other samples in the batch. Fig. 2a shows our framework, where the

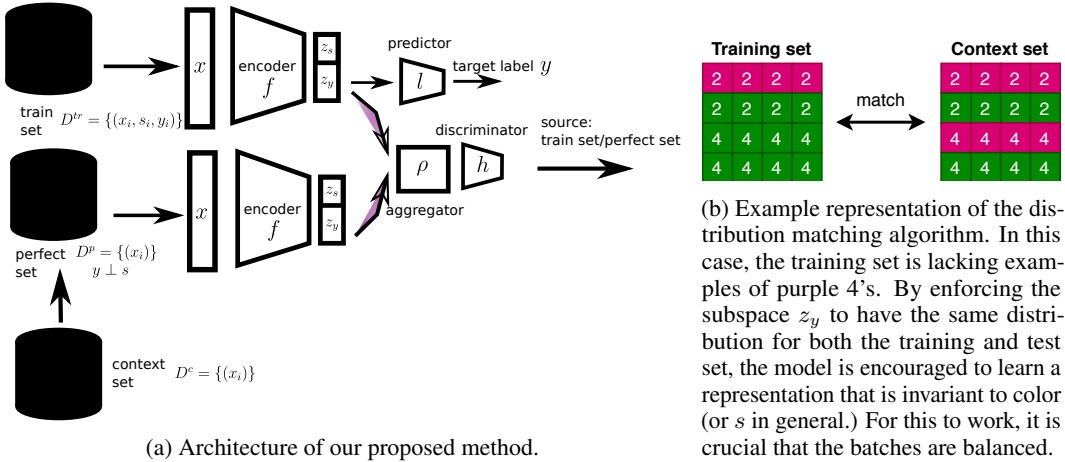

(b) Example representation of the distribution matching algorithm. In this case, the training set is lacking examples of purple 4's. By enforcing the subspace $z_y$ to have the same distribution for both the training and test set, the model is encouraged to learn a representation that is invariant to color (or $s$ in general.) For this to work, it is crucial that the batches are balanced.

(a) Architecture of our proposed method.

Figure 2: Overview of the disentangling framework with a perfect dataset as an inductive bias.

training signal comes from the perfect dataset. Formally, given the training set, $\mathcal{D}^{tr}$ and samples from the *balanced* (i.e. *perfect* – see section 2.2 for how this details on how this can be practically achieved) context set $\mathcal{X}_{perf}$, our learning objective can be written as:

$$L_{match} = \sum_{x \in \mathcal{X}_{tr} \bigcup \mathcal{X}_{perf}} L_{recon}(x, g(z_s, z_y)) + \lambda_1 \sum_{x \in \mathcal{X}_{tr}} L_{sup}(y, l(z_y)) +$$
$$+ \lambda_2 \left( \log h(f(z_y \subset \mathcal{X}_{perf})) + \log(1 - h(f(z_y \subset \mathcal{X}_{tr}))) \right), \quad (8)$$

where $L_{recon}$ and $L_{sup}$ denote the reconstruction loss, and supervised loss, respectively, and $\lambda_1$ and $\lambda_2$ are pre-factors. In practice, this objective is computed over mini-batches, $B$, and the discriminator $h$ is trained via the standard JSD loss (Goodfellow et al., 2014) to map a batch of data points from the training set and the context set to a binary label: 1 if the batch is judged to have been sampled from the context set, 0 otherwise. Its goal is to effectively estimate the probability that a batch of samples, as a set, has been sampled from one distribution or the other. Since the task is a set-prediction one, we require that the function it defines respects the exchangeability of the batch dimension – that is the discriminator's predictions should take into account dependencies between samples in a batch but should be invariant to the order in which they appear, i.e. we have $h(\{z_y\}_{b=i}^B) = h(\{z_y\}_{b=\pi(i)}^B)$ for all permutations $\pi \in \Pi$. To render the entirety of the function $h$ composed of sub-functions $h_1(h_2(h_3...)))$, it requires only the innermost, sub-function, $\rho$ in the chain to have this property. While there are a number of choices when it comes to defining $\rho$, we choose a weighted average $\rho = \frac{1}{B} \sum_i (\{\text{attention}(z_y)\}_{b=i}^B)$, with weights computed according to a learned attention mechanism. It takes the form of the scaled dot-product attention (Vaswani et al., 2017) $\text{attention}(Q, K, V) := \text{softmax}(QK^T/\sqrt{d})V$, , weighting values (V) according to the similarity between the associated key (K) and query (Q) matrices, as measured by their dot-product. Q, K, and V are used after they have been embedded into linear subspaces by matrix-multiplication with learned weight matrices of dimension $\mathbb{R}^{m \times d}$. We found that defining K and V as $z_y$, and Q as the mean of $z_y$ over $B$, yielded good results and leave it to future work to explore more sophisticated methods. The result of $\rho$ is then processed by a series of fully-connected layers, following the DeepSets (Zaheer et al., 2017) paradigm, which ultimately computes a single prediction for the current batch.

We know that the independence condition $y \perp s$ holds in the perfect set, but not in the training set due to sampling bias. To do well, the discriminator should rely on this knowledge. More concretely, since the context and training set have differing support over $\mathcal{S} \times \mathcal{Y}$, namely $(\mathcal{S}_{tr} \times Y_{tr}) \subsetneq (\mathcal{S}_{perf} \times \mathcal{Y}_{perf})$, that support serves as an indicator of the distribution from which the data has been drawn. The scenarios we consider dictate $\mathcal{Y}_{tr} = \mathcal{Y}_{perf}$, making the disentangling well-posed. However, since we wish to use $\mathcal{S}_{ctx} \times \mathcal{Y}_{ctx} \setminus \mathcal{S}_{tr} \times \mathcal{Y}_{tr}$ as the training signal for the encoder, and not the relative frequency of the target classes, it is important that, like the context set, we weight the samples of the training set such that $p(s_{tr}|y_{tr})p(y_{tr})$ is equal for all $s_{tr}, y_{tr} \in \mathcal{S}_{tr} \times \mathcal{Y}_{tr}$. To guide the network towards the desired solution, we supplement this implicit constraint with the explicit constraint that $z_y$ be predictive of $y$, which we achieve using a linear predictor $l$; whenever we have $\dim(\mathcal{S}) > 1$ (in

our experiments this corresponds to the *partial outcomes* setting) we also impose the same constraint on $z_s$, but with respect to $s$. With these conditions met, to fool the discriminator, the encoder must separate out information pertaining to $S$ into the embedded space $z_s$ not part of the discriminator's input, leaving only unprotected information in $z_y$.

## 2.2 IMPLEMENTATION

Our framework overall entails two steps: 1) a method to construct a perfect dataset from an unlabeled context set, and 2) a method to produce disentangled representations using the perfect dataset.

**Constructing approximately perfect dataset via clustering.** We cluster the data points from the context set into $K = \dim(\mathcal{Y}) \cdot \dim(\mathcal{S})$ number of clusters, i.e. the number of data sources $\mathcal{M}_{y,s}$. We use the k-means clustering algorithm, and a recent method based on rank statistics (Han et al., 2020). The cluster assignments can then be used as the basis for constructing a *perfect dataset* for the subsequent disentangling phase. As a result of clustering, the data points in the context set $\mathcal{D}^{ctx}$ are labeled with cluster assignments $\mathcal{D}^{ctx} = \{(x_i, c_i)\}$, $c_i = C(z_i)$. We balance $\mathcal{D}^{ctx}$ so that all clusters have equal size to form a perfect dataset (see fig. 1), and use it as a supervision signal for the disentangling step.

**Clustering requirements.** We do not need to explicitly name the clusters (i.e. finding the demographic groups and labels that the clusters correspond to is unnecessary). The clustering is needed for drawing an equal number of samples from each cluster, for each batch of data. Thus, constructing the perfect dataset in this way does not require solving the linear assignment problem of cluster-source association. In our experiments, we provide an analysis with unsupervised k-means clustering where we do not use annotations from the training set even for the *known* groups. When clustering with the training labels (such as with the rank statistics approach), we use the information that they provide to ensure samples from the *known* subgroups are clustered together with others with the same label.

## 3 EXPERIMENTS

We conduct experiments using the Colored MNIST (Kim et al., 2019; Arjovsky et al., 2019; Kehrenberg et al., 2020) and Adult Income (Dheeru & Karra Taniskidou, 2017) datasets that are publicly available. To validate the first step of creating the perfect dataset, we compare three approaches: the proposed model with clustering via rank statistics (`ZSF+bal. (ranking)`), with clustering via k-means (`ZSF+bal. (k-means)`), and without balancing, when the context set $\mathcal{D}^{ctx}$ is used directly (`ZSF`); followed by the disentangling step as described. Additionally, we evaluate a variant where the batches are balanced with ground truth labels (`ZSF+bal. (ground truth)`).

To validate the second step of learning fair representation via disentangling, we compare with two other baselines. We train a binary classifier on the labeled training data which simply trained with balanced batches (`CNN` for Colored MNIST and `MLP` for Adult Income). This is essentially what Kamiran & Calders (2012b) proposed, so we refer to it as `Kamiran & Calders`. The second is the fairness without demographics (`FWD`) method (Hashimoto et al., 2018) that learns fair classifier with abstract groups. This is the only fairness-based method that is intended for the setting with invisible demographics.

In Adult Income, in the setting of learning with partial outcomes, i.e. when we observe one-sided outcome for one of the demographics, we compare with one additional fairness-aware baseline. It is an adaptation of our model based on a common fair representation learning paradigm (Edwards & Storkey, 2015; Madras et al., 2018; Creager et al., 2019). Using the same AutoEncoder model we do for ZSF to generate a bipartite space, we train an adversarial network to minimize the approximate mutual information between $s_{tr}$ and the representation the $z_y$, with the reconstruction likewise a function of $z_s$ and $z_y$. We refer to this model as `MIM` (Mutual Information Minimization). `MIM` is similar to the FFVAE model proposed Creager et al. (2019), in the sense of learning of a bipartite latent space. However, since our experiments consider only single protected attribute, the disentangling term is irrelevant. Furthermore, rather than encouraging $z_s$ to be predictive of $s$, we encourage $z_y$ to be *un*predictive of $s$ for `MIM`, preventing the possibility of $s$-related information occupying both subspaces. Overall, this adaptation amounts to just modifying the adversarial loss but is only applicable in the setting where both protected groups are present in the training data (*partial outcome setting*), and cannot handle cases in which demographics are missing entirely.

Table 1: Results on Colored MNIST dataset for the task of *two* versus *four* digits classification with a binary protected attribute (purple, green). We consider the scenarios of learning with partial outcomes (30 repeats) and missing demographic (30 repeats). The fairness measures are: the ratio of acceptance rates (AR ratio), true positive rates (TPR ratio), and true negative rates (TNR ratio) between the two demographics, respectively (the closer to one the fairer).

Learning with partial outcomes, the source $\mathcal{M}_{y=\text{'four'},s=purple}$ is invisible.

| | Cluster. Acc. ↑ | Acc. ↑ | AR ratio → 1.0 ← | TPR ratio → 1.0 ← | TNR ratio → 1.0 ← |
|---|---|---|---|---|---|
| ZSF | N/A | $0.829 \pm 0.083$ | $0.306 \pm 0.337$ | $0.31 \pm 0.342$ | $0.994 \pm 0.006$ |
| ZSF+bal. (ranking) | $0.983 \pm 0.006$ | $0.912 \pm 0.056$ | $0.653 \pm 0.232$ | $0.652 \pm 0.227$ | $0.995 \pm 0.005$ |
| ZSF+bal. (k-means) | $0.732 \pm 0.157$ | $0.904 \pm 0.076$ | $0.627 \pm 0.308$ | $0.63 \pm 0.313$ | $0.995 \pm 0.004$ |
| ZSF+bal. (ground truth) | N/A | $0.92 \pm 0.058$ | $0.686 \pm 0.245$ | $0.684 \pm 0.241$ | $0.997 \pm 0.003$ |
| Kamiran & Calders (2012) CNN | N/A | $0.756 \pm 0.009$ | $0.001 \pm 0.005$ | $0.001 \pm 0.005$ | $0.994 \pm 0.004$ |
| FWD Hashimoto et al. (2018) | N/A | $0.765 \pm 0.026$ | $0.045 \pm 0.106$ | $0.045 \pm 0.104$ | $0.989 \pm 0.027$ |

Learning with missing demographics ( purple color), two sources $\mathcal{M}_{y=\text{'four'},s=purple}$, $\mathcal{M}_{y=\text{'two'},s=purple}$ are invisible.

| | Cluster. Acc. ↑ | Acc. ↑ | AR ratio → 1.0 ← | TPR ratio → 1.0 ← | TNR ratio → 1.0 ← |
|---|---|---|---|---|---|
| ZSF | N/A | $0.851 \pm 0.143$ | $0.729 \pm 0.257$ | $0.622 \pm 0.405$ | $0.797 \pm 0.261$ |
| ZSF+bal. (ranking) | $0.922 \pm 0.024$ | $0.868 \pm 0.114$ | $0.876 \pm 0.097$ | $0.783 \pm 0.265$ | $0.702 \pm 0.23$ |
| ZSF+bal. (k-means) | $0.716 \pm 0.153$ | $0.796 \pm 0.104$ | $0.688 \pm 0.219$ | $0.709 \pm 0.328$ | $0.492 \pm 0.296$ |
| ZSF+bal. (ground truth) | N/A | $0.888 \pm 0.088$ | $0.854 \pm 0.089$ | $0.852 \pm 0.175$ | $0.714 \pm 0.203$ |
| Kamiran & Calders (2012) CNN | N/A | $0.759 \pm 0.04$ | $0.239 \pm 0.304$ | $0.288 \pm 0.398$ | $0.738 \pm 0.376$ |
| FWD Hashimoto et al. (2018) | N/A | $0.763 \pm 0.041$ | $0.355 \pm 0.302$ | $0.525 \pm 0.454$ | $0.531 \pm 0.432$ |

Table 2: Results on Colored MNIST dataset for a 3-digits-3-colors task, i.e. classification of the digits *two* versus *four* vs *six* with a protected attribute that can take three values (purple, green, blue). We consider the scenario of learning with partial outcomes with four sources missing (30 repeats). The fairness measures are: the Hirschfeld-Gebelein-Renyi correlation (the lower the better), the mean of the pairwise ratio of acceptance rates (AR ratio mean), true positive rates (TPR ratio mean), and true negative rates (TNR ratio mean) across all pairwise combinations (the closer to one the fairer).

Learning with partial outcomes, the sources $\mathcal{M}_{y=\text{'two'},s=green}$, $\mathcal{M}_{y=\text{'two'},s=blue}$, $\mathcal{M}_{y=\text{'four'},s=blue}$ and $\mathcal{M}_{y=\text{'six'},s=green}$ are invisible.

| | Acc. ↑ | HGR corr. ↓ | AR ratio (mean) ↑ | TPR ratio (mean) ↑ | TNR ratio (mean) ↑ |
|---|---|---|---|---|---|
| ZSF | $0.88 \pm 0.08$ | $0.264 \pm 0.169$ | $0.725 \pm 0.15$ | $0.911 \pm 0.118$ | $0.798 \pm 0.201$ |
| FWD Hashimoto et al. (2018) | $0.62 \pm 0.04$ | $0.803 \pm 0.047$ | $0.157 \pm 0.066$ | $0.383 \pm 0.097$ | $0.418 \pm 0.098$ |
| Kamiran & Calders (2012) CNN | $0.63 \pm 0.05$ | $0.784 \pm 0.069$ | $0.208 \pm 0.1$ | $0.471 \pm 0.145$ | $0.359 \pm 0.036$ |

We report the following performance metrics: clustering accuracy on the context set, classification accuracy and fairness metrics of the prediction task on the test set.

**Colored MNIST dataset with 2 digits.** The colored MNIST dataset is a variant of the MNIST dataset in which the digits are colored, and the color simulates the protected attributes of the digit. We study binary classification, digit *two* versus digit *four*, and explore two settings: with one digit-color source invisible (learning with partial outcomes) and two digit-color sources invisible (learning with missing demographics). Specifically, in the first setting we have training data for digit *two* in green and purple colors, but the digit *four* only comes in green color, so the source $\mathcal{M}_{y=\text{'four'},s=purple}$ is invisible. The second setting is learning with missing demographics, where we have training data for both digits in green, but we do not have training data for the demographics of purple color, i.e. two sources $\mathcal{M}_{y=\text{'two'},s=purple}$ and $\mathcal{M}_{y=\text{'four'},s=purple}$) are invisible. At test time and in the context set we observe all possible colored digits combinations. We follow the colorization procedure outlined by Kim et al. (2019), with the mean color values selected so as to be maximally dispersed. The images are symmetrically zero-padded to a size of 32x32. In the 2-digit case, we use 5,339 images in the unlabeled context set, 2,328 training samples, and 2,014 for testing the classifier.

We report the results in Table 1. In both settings, we are able to clearly outperform the baselines. In the partial outcome setting, we can see that balancing the batches (`bal.`), to emulate a perfect dataset, significantly improves performance, not only in terms of accuracy but in all fairness metrics. The results in Table 1 show relatively high variance, but this likely stems from the smallness of the training set and potentially the imbalance in the classes. As Agrawal et al. (2020) observed, high variance is expected with fairness-enforcing methods (see especially Section 3 there).

For the setting with missing demographics, the results show that balancing is still useful, but only marginally so. This makes sense, because in this setting, the network does not have to ride the fine

Table 3: Results on the Adult Income dataset for the binary classification task of predicting whether an individual earns >\$50,000 with a binary protected attribute *gender* (30 repeats). We consider two scenarios: one where an entire demographic is missing (females are missing from the training set) and one where we only partially observe outcomes for one of the demographics (all females earn less than \$50,000). The fairness measures are: AR (Acceptance Rate) ratio, TPR (True Positive Rate) ratio, and TNR (True Negative Rate) ratio.

Learning with partial outcomes, the source $\mathcal{M}_{y=\text{'above \$50,000'}, s=\text{'female'}}$ is invisible.

| | Cluster. Acc. ↑ | Acc. ↑ | AR ratio → 1.0 ← | TPR ratio → 1.0 ← | TNR ratio → 1.0 ← |
|---|---|---|---|---|---|
| ZSF | N/A | $0.691 \pm 0.014$ | $0.313 \pm 0.066$ | $0.347 \pm 0.07$ | $0.859 \pm 0.029$ |
| ZSF+bal. (k-means) | $0.388 \pm 0.038$ | $0.682 \pm 0.027$ | $0.262 \pm 0.093$ | $0.301 \pm 0.109$ | $0.835 \pm 0.042$ |
| ZSF+bal. (ranking) | $0.699 \pm 0.003$ | $0.708 \pm 0.017$ | $0.488 \pm 0.078$ | $0.551 \pm 0.085$ | $0.909 \pm 0.027$ |
| ZSF+bal. (ground truth) | N/A | $0.731 \pm 0.011$ | $0.556 \pm 0.065$ | $0.63 \pm 0.072$ | $0.897 \pm 0.023$ |
| Kamiran & Calders (2012) MLP | N/A | $0.681 \pm 0.009$ | $0.151 \pm 0.023$ | $0.182 \pm 0.026$ | $0.79 \pm 0.033$ |
| FWD Hashimoto et al. (2018) | N/A | $0.68 \pm 0.008$ | $0.15 \pm 0.022$ | $0.18 \pm 0.023$ | $0.791 \pm 0.038$ |
| MIM+bal. (ground truth) | N/A | $0.658 \pm 0.026$ | $0.215 \pm 0.158$ | $0.246 \pm 0.180$ | $0.893 \pm 0.026$ |

Learning with missing demographics (Females), two sources $\mathcal{M}_{y=\text{'above \$50,000'}, s=\text{'female'}}$, $\mathcal{M}_{y=\text{'below \$50,000'}, s=\text{'female'}}$ are invisible.

| | Cluster. Acc. ↑ | Acc. ↑ | AR ratio → 1.0 ← | TPR ratio → 1.0 ← | TNR ratio → 1.0 ← |
|---|---|---|---|---|---|
| ZSF | N/A | $0.768 \pm 0.024$ | $0.805 \pm 0.124$ | $0.851 \pm 0.131$ | $0.912 \pm 0.035$ |
| ZSF+bal. (k-means) | $0.373 \pm 0.035$ | $0.777 \pm 0.028$ | $0.843 \pm 0.105$ | $0.885 \pm 0.101$ | $0.912 \pm 0.046$ |
| ZSF+bal. (ranking) | $0.606 \pm 0.01$ | $0.775 \pm 0.015$ | $0.801 \pm 0.082$ | $0.868 \pm 0.086$ | $0.911 \pm 0.037$ |
| ZSF+bal. (ground truth) | N/A | $0.767 \pm 0.015$ | $0.81 \pm 0.063$ | $0.872 \pm 0.06$ | $0.923 \pm 0.032$ |
| Kamiran & Calders (2012) MLP | N/A | $0.801 \pm 0.018$ | $0.746 \pm 0.101$ | $0.846 \pm 0.095$ | $0.841 \pm 0.039$ |
| FWD Hashimoto et al. (2018) | N/A | $0.802 \pm 0.018$ | $0.752 \pm 0.102$ | $0.854 \pm 0.097$ | $0.839 \pm 0.038$ |

line of identifying the intended difference between training and context set. Instead, here it is very clear what the difference is: the training set only contains green digits whereas the context set has also purple digits. `k-means` performs relatively poorly here. It might be that it produces batches more biased than random batches which prevents the network from learning the right disentanglement.

**Colored MNIST dataset with 3 digits.** We conduct the experiments for a 3-digits-3-colors variant of ColoredMNIST dataset using the setting of learning with partial outcomes, to investigate how an increase in the number of classes affects disentangling of classes ang groups. We report the results with four sources missing in Table 2 and Table 7 in the Appendix. Our method (ZSF) outperforms the baselines by a significant margin with respect to both accuracy and all fairness metrics. Since, in this case, $\mathcal{S}$ and $\mathcal{Y}$ are both no longer binary, we generalize the fairness metrics applied to the binary datasets in two ways. We compute the mean of the pairwise AR/TPR/TNR ratios across all pairwise combinations. Additionally we compute the minimum (i.e. farthest away from 1) of the pairwise ratios (AR ratio min) as well as the largest difference between the raw AR values (AR diff max) reported in the Appendix. Also we compute the Hirschfeld-Gebelein-Renyi (HGR) maximal correlation (Rényi, 1959) between $S$ and $Y$, serving as a measure of dependence defined between two variables with arbitrary support. See Appendix D for visualizations of the learned fair representation.

**Adult Income dataset.** The Adult Income dataset is a common dataset for evaluating fair machine learning models. In this dataset, each instance is described by $14$ characteristics including gender, education, marital status, number of work hours per week among others, along with a label denoting income level ($\geq$50K or not). We transform the representation into $62$ real and binary features along with the protected attribute $s$. In the whole dataset, 30% of the male individuals earn more than \$50K per year (high income), however of the female individuals only 11% have a high income. Following standard practice in algorithmic fairness e.g. Zemel et al. (2013), we consider gender to be the protected attribute.

For evaluation, we balanced the test set such that all elements of $S \times Y$ are equally sized, observing that a random subset of the data could lead to a majority classifier achieving comparable accuracy to the fairness-unaware baselines, while achieving perfect fairness in terms of TPR ratios. We repeat the procedure with 30 train/context/test splits and report the average performance across repeats. We study the following two settings. 1) *Invisible demographics*: we have training data for males with positive and negative outcomes, but do not have labeled data for females, i.e. $\mathcal{M}_{y=1s=0}$ and $\mathcal{M}_{y=0s=0}$ are missing; 2) *Partial outcomes*: we have labeled training data for males $s = 1$ with both positive and negative outcomes, but for the group of females $s = 0$, we only observe the one-sided negative outcome, so the source $\mathcal{M}_{y=1s=0}$ is invisible. The results are reported in Table 3.

*Partial outcomes*: We see that the proposed approach consistently outperforms the baselines in terms of fairness and performs on-par or better than them in terms of accuracy. The importance of the context-set being properly balanced for our method is reflected in the superiority of `ZSF+bal.` `(ranking)` over other variants, while the k-means variant fails to recognize the correct clustering according to $S \times Y$, and the downstream accuracy suffers. The `MIM` baseline with ground truth balancing performs better than `Kamiran & Calders` and `FWD` in terms of fairness at the expense of accuracy drop. Our `ZSF+bal.` `(ground truth)` variant is dominating `MIM+bal.` `(ground truth)` in terms of both performance metrics.

*Invisible demographics*: In the absence of $s_{tr} =$ Female altogether, `FWD` outperforms the baseline `Kamiran & Calders (2012)` `MLP` in all respects but TNR ratio, being the top-performer in accuracy among the tabulated methods. However, `ZSF+bal.` variants show to be considerably fairer according to all fairness metrics, at the cost of accuracy, and demonstrate the importance of a balanced context set (when compared to `ZSF` alone). In both settings, even though the clustering accuracy is far from perfect, it is good enough to benefit the distribution matching phase, indicating that while $L_{match}$ is sensitive to the quality of $X_{perf}$, it is not overly so.

## 4 DISCUSSION AND CONCLUSION

We have introduced a problem of algorithmic fairness in the presence of *invisible demographics*, which is at the intersection of demographic group fairness with each training data point annotated with protected attributes, and abstract group fairness with unknown protected attributes. We want to protect well-specified demographic groups but some demographics have non-existent labeled training data - those individuals are the invisible demographics. Our proposed model consists of discovering the missing demographic clusters in the unlabeled context set and subsequently learning a disentangled fair representation that can be used at deployment. We consider the train and context sets as coming from the same data domain, such that the knowledge about invisible demographics can be directly transferred from the latter. Extending the model to allow for a domain adaptation step between those sets will be explored in the future. This work is the first attempt in addressing learning with invisible demographics, and we hope it will spark broad interest in the community.

## 5 CURRENT LIMITATIONS

First, dataset consumers should take extra care about the cost-benefit analysis of selecting particular datasets for their machine learning tasks. Although having zero labeled examples for some demographic groups is not uncommon, especially at the intersection of protected attributes, we should do go/no-go decisions w.r.t. this dataset. Corrective action such as fairness interventions or inaction should be recorded.

Second, the problem of fairness has no *one size fits all* solution as fairness definitions are context specific, i.e. different fairness definitions have different meanings in different contexts and not all fairness criteria can be simultaneously fulfilled in one decision process. Every decision process involves different stakeholders, decision makers, individuals affected by the decision, and sometimes the general public, as some decisions (e.g. criminal justice) has impact on the society as a whole. It is also the case that bias by models and perceived bias by a human observer might not be the same and has to be studied in a broad interdisciplinary context.

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

## A  WHY NOT USE A FAIR CLUSTERING METHOD?

Current fair clustering methods (Chierichetti et al., 2017; Backurs et al., 2019; Huang et al., 2019) cluster based on the protected attribute and thus are not applicable to our setting in which the context set is unlabeled and the training set is incomplete with respect to $s$.

## B  DATASET CONSTRUCTION

### B.1  COLORED MNIST BIASING PARAMETERS

To simulate a real-word setting where the data, labeled or otherwise, is usually not naturally balanced, we bias the Colored MNIST training and context sets by downsampling certain color/digit combinations. The proportions of each such combination *retained* in the *partial outcomes* (in which we have one source missing from the training set) and *invisible demographics* (in which we have two sources missing from the training set) are enumerated in Table 4 and 5, respectively. For the 3-digit-3-color variant of the problem, no biasing is applied to either the context set or the training set (the missing combinations are specified in the caption accompanying Table 2); this variant was experimented with only under the partial-outcomes setting.

Table 4: Biasing parameters for the training (left) and context (right) sets of Colored MNIST in the *partial outcomes* setting.

| Combination | Proportion retained | | Combination | Proportion retained |
|---|---|---|---|---|
| (y = 2, s = purple) | 1.0 | | (y = 2, s = purple) | 0.7 |
| (y = 2, s = green) | 0.4 | | (y = 2, s = green) | 0.6 |
| (y = 4, s = purple) | 0.0 | | (y = 4, s = purple) | 0.4 |
| (y = 4, s = green) | 1.0 | | (y = 4, s = green) | 1.0 |

Table 5: Biasing parameters for the training (left) and context (right) sets of Colored MNIST in the *invisible demographics* setting.

| Combination | Proportion retained | | Combination | Proportion retained |
|---|---|---|---|---|
| (y = 2, s = purple) | 0.0 | | (y = 2, s = purple) | 0.7 |
| (y = 2, s = green) | 0.85 | | (y = 2, s = green) | 0.6 |
| (y = 4, s = purple) | 0.0 | | (y = 4, s = purple) | 0.4 |
| (y = 4, s = green) | 1.0 | | (y = 4, s = green) | 1.0 |

### B.2  ADULT INCOME

For the Adult Income dataset, we do not need to apply any synthetic biasing as the dataset naturally contains some bias wrt $s$. Thus, we instantiate the context as just a random subset of the original dataset. Explicit balancing of the test set is needed to yield informative evaluation, however, namely in the penalizing of biased classifiers, but care must be taken in doing so. Balancing the test set such that

$$|\{x \in X | s = 0, y = 0\}| = |\{x \in X | s = 1, y = 0\}| \quad \text{and} \tag{9}$$
$$|\{x \in X | s = 0, y = 1\}| = |\{x \in X | s = 1, y = 1\}|.$$

where for both target classes, $y = 0$ and $y = 1$, the proportions of the groups $s = 0$ and $s = 1$ are made to be the same, is intuitive, yet at the same time precludes sensible comparison of the accuracy/fairness trade-off of the different classifiers. Indeed, with the above conditions, a majority classifier (predicting all 1s or 0s) achieves comparable accuracy to the fairness-unaware baselines, while also yielding perfect fairness, by definition. This observation motivated us to devise an alternative scheme, where we balance the test set according to the following constraints

$$|\{x \in X | s = 0, y = 0\}| = |\{x \in X | s = 0, y = 1\}| \tag{10}$$
$$= |\{x \in X | s = 1, y = 1\}| = |\{x \in X | s = 1, y = 0\}|.$$

That is, all subsets of $\mathcal{S} \times \mathcal{Y}$ are made to be equally sized. Under this new scheme the accuracy of the the majority classifier is 50% for the binary-classification task.

## C   OPTIMIZATION

The hyperparameters and architectures for the AutoEncoder (`AE`), Predictor and Discriminator subnetworks used for the Adult Income and Colored MNIST experiments are detailed in Table 6. For fair comparison, identical hyperparameters are used for the MIM baseline. All networks are trained using the Adam optimizer (Kingma & Ba, 2015).

For ColorMNIST dataset, the baseline `CNN` and `FWD` model use an architecture similar to the encoder with two substitutions. 1) GLU Dauphin et al. (2017) is replaced with Leaky ReLU as the hidden activation; 2) max-pooling is used for spatial downsampling instead of strided convolutions. The final convolutional layer is followed by a global average pooling layer followed by a fully-connected classification layer. For the `MIM` and `ZSF` models, the architecture matches that of the discriminator, excluding the components from the aggregation operation onward in the latter case. All classifiers were trained for 60 epochs with a learning rate of $1 \times 10^{-3}$ and a batch size of 256.

For evaluating on the Adult Income dataset we use scikit-learn's (Pedregosa et al., 2011) logistic regression (`LR`) model, optimized with LBFGS, as the base classifier for K&C, `MIM` and our method (`ZSF`). SVM and LR with cross-validation (`LRCV`) from the same library are also included as baselines. Due to the discrepancy between the training and test sets leading to biased CV estimates, we found that using `LR` with a fixed regularization constant ($C = 1.0$) consistently yielded better performance. For the `FWD` baseline, logistic regression was again used but with it trained via gradient descent (Adam, learning rate=$1 \times 10^{-3}$) instead of via convex optimization due to the non-standard loss function.

Since, by design, we do not have labels for all subgroups the model will be tested on, and bias against these invisible subgroups is what we aim to avoid, properly validating, and thus conducting hyperparameter selection for models generally, is not straightforward. We can use estimates of the mutual information between the learned-representation and $s$ and $y$ (which we wish to minimize wrt to the former, maximize wrt the latter) to guide the process, though as we see from `MIM` baseline, optimizing the model wrt to these metrics obtained from only the training set does not guarantee generalization to the missing subgroups. We can however measure, additionally measure the entropy of the predictions on the encoded test set and seek to maximize it across all samples, or alternatively train a discriminator of the same kind used for training `ZSF` as a measure the shift in the latent space between datasets. We use the latter approach (considering, the learned distance between subspace distributions, accuracy, and reconstruction loss) to inform an extensive grid-search over the hyperparameter space of `ZSF`, and by extension `MIM`, for which we use the same encoder architecture as for `ZSF`, and the same discriminator architecture up until the aggregation step.

For the FWD-model, we allowed access to the labels of the test set for the purpose of hyperparameters selection, performing a grid-search over multiple splits to avoid overfitting to any particular instantiation. Specifically, the threshold ($\eta$) parameter for FWD was determined by a grid-search over the space $\{0.01, 0.1, 0.3, 1.0\}$.

In addition to the losses stated in the distribution matching objective, $L_{match}$, in the main text, we also regularize the encoder by the $\ell^2$ norm of its embedding, finding this to work better than more complex regularization methods such as spectral normalization (Miyato et al., 2018), finding this helped stabilize training. The weight associated with this parameter is denoted as '$\ell^2$-norm weight' Table 6.

## D   QUALITATIVE RESULTS

Given a learned invariant representation, we can generate a reconstruction to visualize the information contained in it. An example of this can be seen in figure 3. This is from our experiment with 3 digits

Table 6: Hyperparameters used for Colored MNIST and Adult dataset experiments.

| | ColorMNIST | Adult |
|---|---|---|
| Input size | $3 \times 32 \times 32$ | 61 |
| **AutoEncoder** | | |
| Levels | 4 | 1 |
| Level depth | 2 | 1 |
| Hidden units / level | $[32, 64, 128, 256]$ | $[61]$ |
| Activation | GLU (Dauphin et al. (2017)) | SELU (Klambauer et al. (2017)) |
| Downsampling op. | Strided Convs. | – |
| Reconstruction loss | MSE | Mixed[1] |
| Learning rate | $1 \times 10^{-3}$ | $1 \times 10^{-3}$ |
| **Clustering** | | |
| Batch size | 256 | 1000 |
| AE pre-training epochs | 150 | 100 |
| Clustering epochs | 100 | 300 |
| Self-supervised loss | Cosine + BCE | Cosine + BCE |
| U (for ranking statistics) | 5 | 3 |
| **Distribution Matching** | | |
| Batch size | 256 | 1000 |
| Training epochs | 250 | 1000 |
| $|z|$ | 128 | 35 |
| $|z_s|$ | 3 | 2 |
| Reconstruction-loss weight | 1 | 1 |
| Predictor weight ($\lambda_1$) | $1 \times 10^{-2}$ | 0.0 |
| $\ell^2$-norm (on encoding) weight | 1 | 0 |
| Adversarial weight ($\lambda_2$) | $1 \times 10^{-3}$ | $1 \times 10^{-2}$ |
| **Predictors** | | |
| Hidden units | - | - |
| Learning rate | $3 \times 10^{-4}$ | $1 \times 10^{-3}$ |
| **Discriminator** | | |
| Hidden units pre-aggregation | $[256, 256]$ | $[32]$ |
| Hidden units post-aggregation | $[256, 256]$ | - |
| Embedding dim (for attention) | 32 | 32 |
| Activation | SELU | SELU |
| Learning rate | $3 \times 10^{-4}$ | $1 \times 10^{-3}$ |
| Updates / AE update | 1 | 1 |

[1]Cross-entropy is used for categorical features, MSE for continuous features.

in Colored MNIST. We can clearly see that the reconstructed invariant representation has lost all color information; instead all digits are magenta-colored, which was the majority color in the training set.

## E    ADDITIONAL METRICS

### E.1    COLORED MNIST DATASET WITH 3-DIGITS-3-COLORS TASK

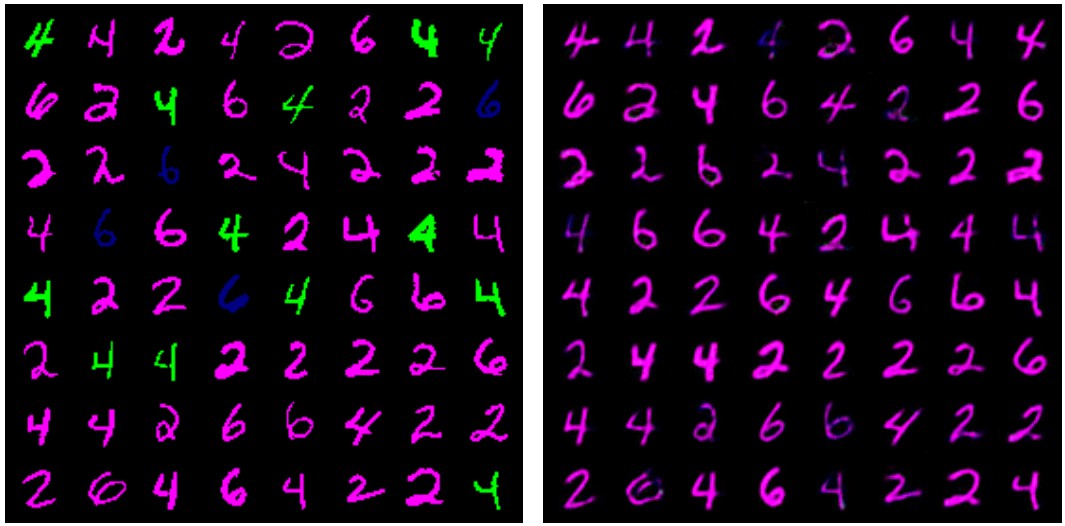

(a) Original images.

(b) Reconstructed invariant representation.

Figure 3: Example images from the experiment with 3 digits (and 3 colors).

Table 7: Results on Colored MNIST dataset for a 3-digits-3-colors task, i.e. classification of the digits *two* versus *four* vs *six* with a protected attribute that can take three values (purple, green, blue). We consider the scenarios of learning with partial outcomes with four sources missing (30 repeats). As the fairness metric, we report the minimum (i.e. farthest away from 1) of the pairwise ratios (AR/TPR/TNR ratio min) as well as the largest difference between the raw values (AR/TPR/TNR diff max) .

Learning with partial outcomes, the sources $\mathcal{M}_{y=\text{'two'},s=green}$, $\mathcal{M}_{y=\text{'two'},s=blue}$, $\mathcal{M}_{y=\text{'four'},s=blue}$ and $\mathcal{M}_{y=\text{'six'},s=green}$ are invisible.

| | AR min. ratio | TPR min. ratio | TNR min. ratio | AR max. diff | TPR max. diff | TNR max. diff |
|---|---|---|---|---|---|---|
| ZSF | $0.604 \pm 0.213$ | $0.866 \pm 0.176$ | $0.702 \pm 0.292$ | $0.236 \pm 0.189$ | $0.133 \pm 0.175$ | $0.297 \pm 0.291$ |
| FWD Hashimoto et al. (2018) | $0.027 \pm 0.05$ | $0.077 \pm 0.145$ | $0.128 \pm 0.147$ | $0.887 \pm 0.101$ | $0.923 \pm 0.145$ | $0.871 \pm 0.147$ |
| Kamiran & Calders (2012) CNN | $0.072 \pm 0.077$ | $0.208 \pm 0.217$ | $0.039 \pm 0.054$ | $0.904 \pm 0.087$ | $0.792 \pm 0.217$ | $0.961 \pm 0.054$ |

