# OpenReview forum: "Zero-shot Fairness with Invisible Demographics"
_ICLR.cc/2021/Conference — Reject_

### Official Review · AnonReviewer3 · 2020-10-27
**Problem proposed has weak and/or problematic motivations, unsure about significance of contributions**

**Rating:** 4
**Confidence:** 3

**Review:**

Summary
- This paper proposed a problem in algorithmic fairness where labeled examples for some demographic groups are completely missing in the training dataset and still the goal is to make predictions that satisfy parity-based fairness constraints.
- The method developed to solve this problem uses a "context" dataset with unlabeled data but containing individuals from all demographics to construct a 'perfect dataset' and 'disentangled representations'

Quality
- This work appears to have only a superficial understanding of the field of algorithmic fairness, hence proposing a problem that in my opinion is artificial. In the case where a dataset has *zero* labeled examples for some demographic groups, this is such an extreme situation that it is a clear red flag that there is a large bias in the collection process and/or the data collection design was poorly done---what is the justification for continuing to use this dataset as is? Is there any real life scenario where one is forced to use this problematic dataset (this could be irresponsible, even unethical), instead of trying to get labels for the "context set" (which is assumed to be available!) or rethinking the data collection process? Clearly one ought to go back to the drawing board in this imagined worst case situation.


References:
  - Kate Crawford. The hidden biases in big data. Harvard Business Review, 1, 2013.
  - Kate Crawford. The trouble with bias. NIPS Keynote https://www.youtube.com/watch?v=fMym_BKWQzk, 2017.
  - Timnit Gebru, Jamie Morgenstern, Briana Vecchione, Jennifer Wortman Vaughan, Hanna Wallach, Hal Daumé III, Kate Crawford. Datasheets for Datasets


- The authors write: "If the model relies only on the incomplete training set, it is not unreasonable to expect that the model to easily misunderstand the invisibles. We can all agree that this sounds unfair, and we would like to rectify this." without any proof or mathematical argument. "this sounds unfair..." is an unrigorous and uncritical statement that doesn't contribute any deeper insight, nor does it engage with existing work on what "unfairness" constitutes. It also does not explain why the paper's method (to massage a clearly problematic dataset) is any less unfair.


The paper also does not cite some related work on algorithmic fairness with missing demographics, e.g.

Recovering from Biased Data: Can Fairness Constraints Improve Accuracy? Avrim Blum∗, Kevin Stangl†


- I'm concerned that the paper calls the missing demographic groups "the invisibles" and then proceeds to still champion the use of the clearly flawed dataset. The algorithm has only intuitive justifications and so does not convince that the missing demographic groups would not be still somehow disadvantaged or find this procedure extremely unjust. The paper does not discuss any of these problematic aspects.



Clarity
- The lack of theoretical guarantees for the algorithm makes it unclear what assumptions are needed for the algorithm to do something meaningful in 'rectifying' the extremely large missing pieces in the original training dataset.

- The explanation for how a "perfect dataset" is constructed is vague (section 2.2). Since the clusters are not explicitly named (i.e. no labels), how is this a "perfect dataset", defined as one where the labels y and group s are independent? Is there any way to check the independence?

---

> ### Author Response · Authors · 2020-11-25
> **Our response**
>
> 1. **Justification for using flawed dataset with invisible demographics**
>
>     Thank you for this important point. The problem of invisible demographics is indeed real. Selective labels problem [1], intersectional fairness [2,3], and a combination of both can easily translate to partial outcomes and missing demographics. The Adult Income dataset has 0 samples with Income >50K of black females at the age of 40 (further detailed for other intersectional groups for this dataset can be found in our comments to **AnonReviewer4)**. This dataset has been used by a number of prior studies in the fairness-aware machine learning literature such as Zemel et al. 2013, Zafar et al. 2017, Madras et al. 2018. We do agree that dataset consumers should take extra care about the cost-benefit analysis of selecting particular datasets for their machine learning tasks. Any corrective action such as fairness interventions or inaction should be recorded. We have added a section on "current limitations" in the revised manuscript.
>
>     [1] H. Lakkaraju, J. Kleinberg, J. Leskovec, J. Ludwig, and S. Mullainathan. The selective labels problem: Evaluating algorithmic predictions in the presence of unobservables. In Proceedings of the 23rd ACM SIGKDD International Conference on Knowledge Discovery and Data Mining, 2017.
>
>     [2] J. Buolamwini and T. Gebru. Gender shades: Intersectional accuracy disparities in commercial gender classification. In Proceedings of the ACM Conference on Fairness, Accountability, and Transparency, 2018.
>
>     [3] M. Kearns, S. Neel, A. Roth, and Z.S. Wu. Preventing fairness gerrymandering: Auditing and learning for subgroup fairness. In Proceedings of the International Conference on Machine Learning, 2018.
>
> 2. **Theoretical results on algorithmic fairness with missing demographics (Blum and Stangl)**
>
>     Blum and Stangl considered two forms of data corruptions: a) under-representation of positive examples in a disadvantaged group, and b) a substantial fraction of positive examples in a disadvantaged group mislabeled as negative. Theoretical results are achieved by assuming equal base rates across groups. Blum and Stangl noted that this assumption may not be realistic in all settings. We do not assume equal base rates (perfect dataset) but we aim to construct a perfect dataset from an unlabeled context set. Ideally, our theoretical results should first bound or characterize the difference of learning with a perfect dataset and learning with an approximately perfect dataset in probabilistic terms. We can subsequently apply the union bound utilising results such as from Blum and Stangl to make a statement about recovering the Bayes Optimal Classifier. We have not managed to do so.
>
> 3. **The explanation for how a "perfect dataset" is constructed is vague (section 2.2). Since the clusters are not explicitly named (i.e. no labels), how is this a "perfect dataset", defined as one where the labels y and group s are independent? Is there any way to check the independence?**
>
>     Given that we are dealing with discrete variables, independence is achieved if we have equal proportions of all combinations, i.e., all combinations are equally represented (P(y,s)=P(y)P(s) ⇒ P(y,s)=0.25 for binary y and s). So if we manage to identify all the clusters that correspond to all the combinations of y and s, then we can sample from these clusters at an equal rate to achieve a balanced dataset in which y and s are not correlated.
>
>     It's true that the clusters are not named, but this is not necessary for this task. To compute the clustering accuracy, we actually have to solve the linear assignment problem of cluster-source association (i.e. we need to explicitly name the clusters). As such, we "name the cluster" only for assessing the quality of the approximate perfect dataset.

---

### Official Review · AnonReviewer2 · 2020-10-27
**Interesting work on zero-shot fairness with partial demographics**

**Rating:** 5
**Confidence:** 4

**Review:**

#Summary

This paper studies zero-shot fairness where the demographic information is partially unavailable, but assuming the existence of a context dataset that contains all labels x all demographics (including the invisible). The paper proposes a disentanglement algorithm that separates information of the label and demographics, under two zero-shot settings: 1) learning with partial outcomes: both labels and both demographics are available, but for one of the demographics only negative outcome is present; 2) learning with missing demographics: one of the demographics is completely missing.

#Pros
- Zero-shot fairness is a very important topic under many practical settings, where the demographic information can be (partially) missing due to sampling bias or privacy reasons.
- The two zero-shot settings presented in this paper are both very interesting, and the paper did a good job decomposing the two scenarios in the methods and experimental section.
- The paper is clearly presented, with careful analysis over each of the proposed component, with proper ablation studies.

#Cons
- The biggest concern I have is the clustering part of the context set into a perfect set. This seems to be a prerequisite for the disentangle algorithm to perform well. However, there is no guarantee over the clustering quality, and this is partially reflected in the experiments (table 1 & 2) as well. For example, while ranking-based clustering achieves reasonable clustering accuracy, k-means seems to be rather bad for certain datasets (e.g., Adult Income). In addition, how does the distribution of the label x demographics on the context dataset affect clustering quality? I can imagine under extreme cases, if the distribution is very skewed (some of the label x demographic has very scarce data), then it is hard to get good clusters, which is very likely to happen in practice if the training distribution is already skewed.
I think some further analysis on this is required, e.g., how the cluster quality differs w.r.t. different retained proportions of each quadrant.

- The experimental results seem to present different trade-offs for the proposed approaches. There doesn't seem to be a single algorithm that has a clear better performance compared to the baselines. E.g., for colored-MNIST ZSF seems to work a bit better, but for Adult Income MIM+bal, and FWD seem to work better. The performance also varies a lot across different fairness metrics as well.

- Although the topic of zero-shot fairness is very important, the end-to-end setting in this paper is a bit artificial. It requires two things, 1) both label $y$ and demographic $s$ are present in the training data, although some of the quadrants are allowed to be missing; 2) there exists a context set that has all quadrants available for $y$ and $s$, thus can be used for balancing and learning the disentangled representations. I wonder how realistic this setting is in practice. It is very likely that 1) is true in real-world but the requirement of 2) makes the setting a bit constrained, what if during deployment time, no context set is available and online inference (for each incoming individual) is needed? Or, what if I have a context set, but some of the quadrants are also missing, and even worse, the missing quadrants are different from the ones missing in training?

#Over recommendation

I think this paper studies a very interesting problem but some further analysis, e.g., how the distribution over the context data affects the results, and how to make the algorithm work reliably better in practice, is needed. Overall I think this is a borderline paper.


#Minor comments and questions
- In the experiments, for colored-MNIST, a comparable portion for each quadrant is retained for the context dataset, have you tried different retained portions and how does that affect clustering quality? Have you tried some of the more extreme settings (e.g., more skewed distribution over |S|x|Y|) and will you still obtain reasonable clusters?
- I didn't find how the context dataset is constructed for Adult Income, could you provide more information on this?
- How are $\lambda_1$ and $\lambda_2$ chosen in the experiments?
- Some of the error bars in table 1&2 are rather large, could the authors further clarify which set of the results are statistically significant?

---

> ### Author Response · Authors · 2020-11-25
> **Our response**
>
> 1. **There doesn't seem to be a single algorithm that has a clear better performance compared to the baselines. E.g., for colored-MNIST ZSF seems to work a bit better, but for Adult Income MIM+bal, and FWD seem to work better.**
>
>     Experiments for the Adult Income dataset have been redone using improved hyperparameters and corrected evaluation protocol, the error being that the weighted-sampling described in Section 2.1 not had not been used for training of the classifier for either our method or the baselines. Please refer to Table 3 for an updated version.
>
> 2. **What if during deployment time, no context set is available and online inference (for each incoming individual) is needed? Or, what if I have a context set, but some of the quadrants are also missing, and even worse, the missing quadrants are different from the ones missing in training?**
>
>     Our context set is much like the deployment dataset. If a context set is not available, we should  consider a transductive learning setting where the deployment set is our context set. It is well-known that an online setting is strictly harder than a batch setting. In the future version, we will work to extend our zero-shot fairness framework for a transductive online learning setting of Ben-David, Kushilevitz, and Mansour [1].
>
>     [1] S. Ben-David, E. Kushilevitz, and Y. Mansour. Online learning versus offline learning. Machine Learning 29:45-63, 1997.
>
> 3. **Minor comments and questions**
>
>     3.1 In the experiments, for colored-MNIST, a comparable portion for each quadrant is retained for the context dataset, have you tried different retained portions and how does that affect clustering quality? Have you tried some of the more extreme settings (e.g., more skewed distribution over $|S|\times|Y|$) and will you still obtain reasonable clusters?
>
>     Exploring the effect of the size of the context set on model performance is something we are keen to explore in order to test the limits of the model.  As mentioned above, we are also interested in exploring the extreme case where we do not have a context set at all and must resort to transductive learning in which the distribution of the training set is matched directly to that of the test/deployment set.
>
>     3.2 I didn't find how the context dataset is constructed for Adult Income, could you provide more information on this?
>
>     For the Adult Income dataset, the context set is simply a regular subset of the data, which unlike Colored MNIST, is naturally biased with respect to the protected attribute, Gender; we have updated the manuscript (Appendix B.2, specifically) to include this detail.
>
>     3.3 How are $\lambda_1$ and $\lambda_2$ chosen in the experiments?
>
>     As detailed in the Table 5 of the Appendix, $\lambda_1$ is 10^-2 and $\lambda_2$ is 10^-3 on ColorMNIST; $\lambda_1$ is 0 and $\lambda_2$ is 10^-2 on Adult. We also elaborate on the hyper-parameter tuning in the section C of the Appendix.
>
>     3.4 Some of the error bars in table 1&2 are rather large, could the authors further clarify which set of the results are statistically significant?
>
>     Please refer to our answer to **AnonReviewer4.**

---

### Official Review · AnonReviewer1 · 2020-10-27
**Well organized paper on a relevant problem, but lacking in key experiment details.**

**Rating:** 6
**Confidence:** 5

**Review:**

############# Summary of contributions ##############

This paper introduces the problem of enforcing group-based fairness for “invisible demographics,” which they define to be demographic categories that are not present in the training dataset. They assume access to a “context set,” which is an additional unlabeled dataset that does contain the invisible demographic categories of interest. They further provide an algorithm for enforcing fairness on these invisible demographics using this context set.

Specifically, their contributions are:

- Algorithmic: They provide an algorithm for enforcing fairness on these invisible demographics. This algorithm involves first applying clustering methods on the context set to “balance” it, followed by disentangled representation learning and on the “balanced” context set.

- Empirical: They provide experiments on two benchmark datasets (colored MNIST and Adult) comparing their proposed method to multiple baselines.

############# Strengths ##############

- The paper is organized well, and the problem of “invisible demographics” is described and motivated well using concrete examples.

- The architecture of the proposed method is documented clearly in Figure 2.

- Their architecture builds on state of the art techniques such as DeepSets (Zaheer et al. 2017). Using DeepSets, the discriminator in their architecture estimates the probability that a given batch of samples, as a set, has been sampled from one distribution or the other. Preserving the set invariance to permutations is useful here, and different from a typical GAN discriminator.

- The baselines in the experiment section are thorough. It’s useful to see a comparison between their clustering + balancing + disentangling method and the baseline methods of ZSF, which has balancing + disentangling but no clustering, and ZSF + bal. (ground truth), which has ground truth clusters + balancing + disentangling.

############# Weaknesses ##############

- The experiments section does not describe the implementation of the comparison to Hashimoto et al. 2018. Notably, the methodology of Hashimoto et al. 2018 is not specifically meant to enforce equality of acceptance rates, true positive rates, or true negative rates -- it only minimizes the worst case loss over unknown demographics.

- The authors do not provide any description of hyperparameters tuned, or any use of a validation set for hyperparameter tuning. I could not find this in the appendix either. In fact, on page 7, they say that they “repeat the procedure with five different train/context/test splits”, which suggests no validation set. The parameters for the clustering methods are not given, and I find it hard to believe that no hyperparameters were tuned. Can the authors specifically provide the hyperparameters used, whether/how they were tuned, and any validation methods used (whether it be a validation set or cross validation)?

- The experiments are all done with binary protected groups: purple vs. green for the colored MNIST dataset, and male vs. female for the Adult dataset. Furthermore, these groups are not hugely imbalanced in the context set to begin with. This makes the clustering task easier. It would be interesting to see experiments with protected groups with more than two categories. For example, in the Adult dataset, the race feature is highly inbalanced, with a very small proportion of examples labeled as Asian-Pac-Islander or Amer-Indian-Eskimo. It would be interesting to see how the clustering techniques compare when the context set includes more than two protected categories, there is initial strong data imbalance between those groups, and the “invisible demographic” has relatively few data examples in the context set. This may not be entirely necessary for acceptance this round, but could be an interesting future experiment.

The notation is in multiple cases unclear/inconsistent, possibly due to typos. Examples listed below:

- In the last paragraph on page 5, the notation and description of the support is confusing and not well defined. First, \mathcal{S} and \mathcal{Y} are themselves sets as defined in Section 2.1. Can the authors more specifically define what they mean by Sup(\mathcal{Y}_tr)? Is this the set of elements from \mathcal{Y} that are contained in the training set? If so, why not just notate this as \mathcal{Y}_tr alone? The additional “Sup” notation is confusing and appears unnecessary. Furthermore, what do the authors mean when they say, “we wish to use Sup(\mathcal{S}_{ctx} \times \mathcal{Y}_{ctx}) \ Sup(\mathcal{S}_{tr} \times \mathcal{Y}_{tr}) as the training signal for the encoder”?

- [Top of page 6: “whenever we have |S| > 1”] -- What does this notation mean? Is this the absolute value of the random variable S? This doesn’t quite make sense given that S was previously stated to be a discrete-valued protected attribute, which could be a vector with p entries. The next statement of this corresponding to the “partial outcomes” setting is thus also unclear.

- [Section 2.2: “c_i = C(z_i)”] -- What is z_i here? Is z_i the vector of (z_s, z_y) for the input features x_i?

############# Recommendation ##############

UPDATE (after author response): I appreciate the authors' response. The inclusion of the hyperparameters are helpful. I also think it's an improvement that the authors added a comparison to ZSF+bal.(ground truth) to the Adult experiment.

I still have a question about the experimental comparison to Hashimoto et al. (called "FWD" in this paper). Is the version of "FWD" implemented in this paper using exactly the same fairness criterion as in the Hashimoto et al. paper? If so, am I correct in saying that the "FWD" comparison in the experiments section does not directly constrain for any of the measured AR ratio, TPR ratio, or TNR ratio? The authors should clarify this in a later version.

Overall, I'm willing to raise my score to a 6, but still think the paper is borderline. The paper could still use some improvement in covering related work on the problem of fairness where the protected attributes are not fully known (including the references I suggested).

------------- OLDER RECOMMENDATION BELOW -------------

Overall, my recommendation is 5: Marginally below acceptance threshold. The paper states an interesting and practically relevant problem of enforcing fairness with “invisible demographics.” The methodology is overall well documented, and the experimental baselines make sense. However, the implementation detail in the experiments section is severely lacking, including description of hyperparameters/validation methods and implementation details for the comparison to Hashimoto et al. If the authors provide some of these details and answer some of my notation questions, then I would be willing to raise my score.

############# Questions and clarifications ##############

- Why is there no comparison to ZSF+bal. (ground truth) on the Adult dataset?

- Can the authors clarify what the ZSF alone baseline is doing in the experiments section? It’s not written super clearly in the text. Does ZSF alone simply replace the perfect set in Figure 2 with the context set?

############# Additional feedback ##############

- Below I’ve listed some additional related work in the setting where protected attributes are unknown. This is not factored into the review, as these settings seem different enough and some of these works are recent.

Lamy et al. Noise-tolerant fair classification. NeurIPS, 2019.

Awasthi et al. Equalized odds postprocessing under imperfect group information. ICML, 2020.

Wang et al. Robust Optimization for Fairness with Noisy Protected Groups. arXiv:2002.09343, 2020

- [page 3: “We can all agree that this sounds unfair”] -- nit: this wording seems unnecessarily strong to me. Let’s not claim that “we would all agree” on something, especially when the meaning of unfair has not yet been defined.

- [page 5]: There appear to be multiple typos in the paragraph following equation (10), where the variables V, Q, K are not written in math mode, and are instead just capital letters in the text.

---

> ### Author Response · Authors · 2020-11-25
> **Our response**
>
> 1. **The implementation detail in the experiments section is severely lacking, including description of hyperparameters/validation methods and implementation details for the comparison to Hashimoto et al.**
>
>     We apologize for these omissions and have since incorporated them into the Appendix C. Table 5 contains a full specification of the hyperparameters used in the training of our ZSF model and details regarding the baselines are described textually. We will also provide the reviewers with a link to an anonymous GitHub repository containing our code and the scripts needed to reproduce the experiments in the paper.
>
> 2. **Answer some of the notation questions.**
>
>     Thank you for pointing the notational inconsistencies out in Section 2; we have amended the notation according to your feedback. To answer the questions raised about this:
>
>     - $Sup(\mathcal{Y}^tr)$ was intended to denote all values of the class label, y, present in the labelled training set.
>     - By "we wish to use $Sup(\mathcal{S}^{ctx} \times \mathcal{Y}^{ctx}) \ Sup(\mathcal{S}^{tr} \times \mathcal{Y}^{tr})$ as the training signal for the encoder" we mean that since the discriminator can determine the origin of a batch of ($z_y$) embeddings (whether it came form a sample from the training or context set) by inferring its support over $S \times Y$ (with both dataset containing all possible values of Y), to succeed in the minimax game, the encoder must learn to properly partition the s-related and s-unrelated information into $z_s$ and $z_y$ respectively. For instance, if the discriminator can determine that a batch contains purple 4s, when there are none in the training set, then it can safely conclude that the batch in question is from the context set (and vice-versa) and the encoder should take action to avoid this by removing color-information from $z_y$.
>     - By |S| we wished to denote the cardinality of the set of possible s-labels in the training set, and so the full statement should be interpreted as "whenever we have more than a single demographic (defined by S) in our labelled dataset". We have replaced this notation with dim() for the sake of clarity.
>     - The $z_i$ in $c_i = C(z_i)$  is distinct from the z mentioned in the disentanglement step - while for both this one and the preliminary clustering step, an autoencoder is used for learning an embedding, in the latter case there is no splitting of it into $z_y$ and $z_s$. The aforementioned equation simply means for each data-point, $x_i$, we encode it using an autoencoder (which is not shared between steps) before feeding it to the clusterer C to produce a cluster assignment $c_i$.
>
> 3. **Questions and clarifications**
>
>     3.1 **Why is there no comparison to ZSF+bal. (ground truth) on the Adult dataset?**
>
>     We are aware that our results for the Adult Income dataset were lacking in the initial version of the manuscript. These results have since been redone and have been incorporated into the updated version and now correctly include the ZSF-with-ground-truth balancing baseline.
>
>     3.2 **Can the authors clarify what the ZSF alone baseline is doing in the experiments section? It’s not written super clearly in the text. Does ZSF alone simply replace the perfect set in Figure 2 with the context set?**
>
>     Yes, ZSF alone simply replaces the perfect set with the context set.

---

### Official Review · AnonReviewer4 · 2020-10-28
**Suspicious experimental results and unclear merit**

**Rating:** 5
**Confidence:** 3

**Review:**

This paper tackles a fair classification problem with an invisible demographic, a situation where the records who have some specific target labels and sensitive attributes are missing. In this setting, the authors introduce a disentangled representation learning framework to make the resultant classifier fair by taking advantage of the additional dataset, context dataset. They demonstrate by the empirical evaluations that the proposed disentangled representation learning algorithm success to mitigate unfair bias by utilizing the perfect dataset, a dataset in which the target label and sensitive attribute are independent. Usually, the perfect dataset is unavailable; hence, they introduce a method to convert the context dataset into the perfect dataset. The authors also show that even if the context dataset is not perfect, the presented method successes to mitigate an unfair bias.

The strong points of this paper are as follows:
- This paper introduces a potentially interesting problem, the invisible demographic.

The weak points of this paper are as follows:
- The experimental results have a high variance. Hence, they are weak to support the significance of the proposed algorithm.
- The motivation of the proposed method is unclear. Some existing methods already solve most of the crucial situations considered in this paper.
- This paper lacks a comparison with the important related method.
- Presentation is poor. I cannot follow the description of the algorithm.

My recommendation is rejection. The main reason is that I have concerns about suspicious behavior in the experimental results. Also, the proposed method is not well-motivated, and its merit is unclear.

I am very suspicious about the experimental results. The standard deviations for the fairness metrics shown in Table 1 and Table 2 are considerably high. Why can we believe the successful mitigation of unfair bias of the proposed method from these results? Even if I believe the reported values, due to the large standard deviation, we cannot say the authors' method outperforms the others but can only say it is competitive.  I don't think this is a significant result.

Parts of the invisible demographic problem are already solved. For example, a situation where records in some classes are missing is solved by utilizing semi-supervised learning techniques, e.g., Hsieh et al. Classification from Positive, Unlabeled and Biased Negative Data. In ICML'19. For a situation where the sensitive attributes are missing, there are several works, including
- N. Kallus et al. Residual Unfairness in Fair Machine Learning from Prejudiced Data. In ICML'18.
- A. Coston et al. Fair Transfer Learning with Missing Protected Attributes. In AIES'19.
It is a rare situation where records with a specific combination of the target class and demographic group are missing. These existing methods already solve other cases. Therefore, it is unclear that the proposed method has merits compared to the existing ones.

There is a fair classification method based on disentangled representation learning:
- E. Creager et al. Flexibly Fair Representation Learning by Disentanglement. In ICML'19.
Because this method and any fair classification methods can apply to the problem tackled by this paper, it is necessary to compare the proposed method with them. I know these methods are not designed to work in the invisible demographic situation; however, it is unclear if they do not work in the situation without empirically evaluating them.

I cannot understand the introduced objective function in Eq. 10. What is the meaning of $f(z_y \subset \mathcal{X}_{perf})$ and $f(z_y \subset \mathcal{X}_{tr})$? While the function $f$ takes $x$ as its input, it takes a boolean value in Eq. 10.

What is clustering accuracy? Its definition is missing.

### Minor comments
- While I understand the situation where the whole sensitive attributes are missing, I wonder if it is a realistic situation that a part of the target label and sensitive attributes are missing. Is there a concrete dataset that invisible demographic situation occurs?
- I cannot make sure about the notation of $\mathcal{M}_{y=1,s=0}=\emptyset$. If my understanding is correct, the set $\mathcal{M}$ (omit subscript $y=1,s=0$ because it doesn't work) comprise of whole data points whose target label and sensitive attribute are $y=1$ and $s=0$, respectively. It involves not only the target data points but also unobserved data points available in the world. From this perspective, $\mathcal{M}=\emptyset$ means that there are no people whose target label and sensitive label are 1 and 0, respectively, in the world. In this case, we cannot construct the context and deployment sets that satisfy Eq. 3 or Eq. 4.
- Typo on page 3, first paragraph:   but in in contrast to ->  but in contrast to

---

> ### Author Response · Authors · 2020-11-25
> **Our response**
>
> 1. **Suspicious experimental results due to high variance of the fairness metrics**
>
>     Prior work by Agrawal et al (2020) has pointed out that group-fairness metrics  incur higher variance compared with accuracy due to stochasticity in the train-test splits and optimization process. In the case of Colored MNIST, the high variance can also be chalked up to the small size of the labelled dataset (60% of 10% of the total MNIST training data) following subsampling and its division into context and training sets.  Our results on the Adult dataset do not show such a high variance. A different splitting procedure might ameliorate some of this.
>
>     [1] Agrawal A, Pfisterer F, Bischl B, Chen J, Sood S, Shah S, Buet-Golfouse F, Mateen BA, Vollmer S. Debiasing classifiers: is reality at variance with expectation?. arXiv preprint arXiv:2011.02407. 2020 Nov 4.
>
> 2. **How realistic is a part of the target label and sensitive attributes are missing (our learning with partial outcomes scenario)**
>
>   When analyzing a train set of the Adult Income dataset, one of the most common datasets for fairness analysis, we found that it:
>
>   Has 0 samples of native-country_Holand-Netherlands and Income >50K
>   Has 0 samples of native-country_Outlying-US(Guam-USVI-etc) and income >50K
>   Has 0 samples with Income >50K of black females at the age of 40 (in contrast, there are 44 samples of white females, and 202 white males, 7 black males, with Income >50K)
>
>   These exemplify our setting with partial outcomes, with the sensitive attributes related to native country, race, age and gender.
>
> 3. **Lacks of comparison with N. Kallus et al. [Residual Unfairness in Fair Machine Learning from Prejudiced Data. In ICML'18], A. Coston et al. [Fair Transfer Learning with Missing Protected Attributes. In AIES'19], Creager et al. [Flexibly Fair Representation Learning by Disentanglement. In ICML'19].**
>
>     Thank you for the comments. We agree that it would be advantageous to place into the right perspective the contributions and relations to those previous works. In residual unfairness/selective label, that has been highlighted, the problem comes from the fact that there is a difference between the decision taken place in real life, and the predictions that the machine learning system is trained to perform. For example: In the bank loan application scenario, the decision is whether or not to give a loan, where as the ML prediction is whether or not the applicant will pay back the loan. Importantly, the ML is trained on historic data of the applicants that did/did not pay back the loan, meaning they have got a loan in the first place. So this means, that if a person has never got a loan, the associated prediction in ML will most likely be ‘not able to pay back the loan’, as the only people who can pay back the loan are those that got a loan in the first place. This is in corresponded with our setting of learning with partial outcomes, where we only observe one-sided decisions w.r.t. certain protected characteristics - if the persons with certain protected characteristics has never got a loan / were always rejected, the ML system will ignore positive outcomes for those individuals.
>
>     As regards Creager et al., 2019, the pre-existing MIM baseline does closely-resemble  the FFVAE model proposed therein, with the key distinctions being
>
>     1.  we do not apply a disentanglement loss to the subspace associated with the protected attribute. Since we only have a single protected attribute in our setups, enforcing disentanglement between the different factors of z_s is irrelevant (calibrating the fairness of predictions by composition of subspaces, each associated with a different sensitive attribute, being the focus of Creager et al., 2019);
>     2. an adversary is used to expel information related to s from z_y; Creager et al., 2019 takes the opposite approach of having a classifier predict s from z_s
>
>     An abbreviated discussion of this kind has been appended to the explanation of the MIM baseline given in the main text. While the FFVAE model may not be entirely suitable, we do think that a baseline which encourages disentanglement of the entire latent space (a property shown to implicitly promote fairness when sensitive attributes are unobserved; Locatello et al., 2019), rather than just over a subset of it, would absolutely be worth having.
>
>     [1] Locatello F, Abbati G, Rainforth T, Bauer S, Schölkopf B, Bachem O. On the fairness of disentangled representations. In Advances in Neural Information Processing Systems 2019 (pp. 14611-14624).

---

### Author Response · Authors · 2020-11-25
**Summary of Changes**

We have updated our manuscript with five principal changes:

1. We have added results for a 3-digit-3-color variant of Colored MNIST, under the partial-outcome setting, to the main text (Table 2), noting that our method (ZSF) outperforms the baselines by a significant margin with respect to both accuracy and all fairness metrics. We visualize the invariant representations in the appendix. Since, in this case, S and Y are both no longer binary, we generalize the fairness metrics applied to the binary S/Y datasets in two ways:
    1. We compute the mean of the pairwise AR/TPR/TNR ratios. In  the appendix, we additionally report the minimum (i.e. farthest away from 1) of the pairwise ratios (min. ratio) as well as the largest difference between the raw values (max. diff).
    2. We compute the Hirschfeld-Gebelein-Renyi (HGR) maximal correlation between S and $\hat{Y}$, serving as a measure of dependence defined between two variables with arbitrary support.
2. The means and standard deviations for all results are now computed over 30 random seeds (note that it's not the standard error, but the standard deviation.)
3. Experiments for the Adult Income dataset have been redone using improved hyperparameters and corrected evaluation protocol, the error being that the weighted-sampling described in Section 2.1 had not been used for training of the classifier for either our method or the baselines. The results now also include a ZSF-with-ground-truth-balancing baseline that  Reviewer 1 noted was previously missing.
4. We have updated Appendix C to include a table of the full set of hyperparameters used for the clustering and distribution-matching phases of the algorithm for both Colored MNIST and the Adult Income dataset, as well as an explanation of how both these hyperparameters and those of the baselines (including FWD) were determined.
5. A short discussion of the current limitations of the work, including some caveats about when it is appropriate to use our method, and about algorithmic fairness in general.

---

### Decision · Program_Chairs · 2021-01-07
**Final Decision**

**Decision:**

Reject

**Comment:**

The paper studies the problem of satisfying group-based fairness constraints in the situation where some demographics are not available in the training dataset. The paper proposes to disentangle the predictions from the demographic groups using adversarial distribution-matching on a "perfect batch" generated by a clustered context set.

Pros:
- The problem of satisfying statistical notions of fairness under "invisible demographics" is a new and well-motivated problem.
- Creative use of recent works such as DeepSets and GANs applied to the fairness problem.

Cons:
- Makes a strong assumption that the clustering of the context set will result in a partitioning that has information about the demographics. This requires at the very least a well-behaved embedding of the data w.r.t. the demographic groups, and a well-tuned clustering algorithm (where optimal tuning is difficult in practice on unsupervised problems) -- but at any rate, as presented, the requirements for a "perfect batch" is neither clear nor formalized.
- Lack of theoretical guarantees.
- Various concerns in the experimental results (i.e. proposed method does not clearly outperform other baselines, high variance in experimental results, and other clarifications).

Overall, the reviewers agreed the studied problem is new, interesting and relevant to algorithmic fairness; however, there were numerous concerns (see above) which were key reasons for rejection.